# Landslide Deformation Prediction Based on a GNSS Time Series Analysis and Recurrent Neural Network Model

**Jing Wang** [1], **Guigen Nie** [1,2,*] , **Shengjun Gao** [3], **Shuguang Wu** [1], **Haiyang Li** [1] **and Xiaobing Ren** [1]

1   GNSS Research Center, Wuhan University, Wuhan 430079, China; jingwangwhu@whu.edu.cn (J.W.); shgwu@whu.edu.cn (S.W.); haiyangli@whu.edu.cn (H.L.); xiaobinren@whu.edu.cn (X.R.)
2   Collaborative Innovation Center for Geospatial Information Technology, Wuhan 430072, China
3   Chinese Antarctic Center of Surveying and Mapping, Wuhan 430079, China; sjgao@whu.edu.cn
*   Correspondence: ggnie@whu.edu.cn

**Abstract:** The prediction of landslide displacement is a challenging and essential task. It is thus very important to choose a suitable displacement prediction model. This paper develops a novel Attention Mechanism with Long Short Time Memory Neural Network (AMLSTM NN) model based on Complete Ensemble Empirical Mode Decomposition with Adaptive Noise (CEEMDAN) landslide displacement prediction. The CEEMDAN method is implemented to ingest landslide Global Navigation Satellite System (GNSS) time series. The AMLSTM algorithm is then used to realize prediction work, jointly with multiple impact factors. The Baishuihe landslide is adopted to illustrate the capabilities of the model. The results show that the CEEMDAN-AMLSTM model achieves competitive accuracy and has significant potential for landslide displacement prediction.

**Keywords:** GNSS time series analysis; landslide displacement prediction; attention mechanism; deep learning

## 1. Introduction

Landslide disaster is one of the crucial topics in geological research [1]. The sustainable development of economies and society is seriously threatened as a result of landslide disasters [2]. Reliable early warning systems are a reasonable approach for landslide risk reduction [3,4]. The mechanisms analysis and prediction of landslide movements are the key components of landslide early warning [5–7]. Therefore, it is judicious to carry out landslide displacement prediction.

Landslide displacement prediction models can be divided into two categories: physical models and numerical models [8,9]. Traditional physical models provide a physical explanation for the prediction work according to geological theory [10]. Saito established a three-stage theory of landslide creep failure in 1968 [11,12], and Hoek proposed the extension line method to predict the time-displacement curve of Chilean landslides in 1977 [13]. However, physical models are deficient in their ability to meet the demands of dynamic large landslide prediction [14–16]. With the rapid development of mathematical statistical theory and intelligent algorithms, numerical models have become more popular [5]. Numerical models fully consider the complexity and nonlinearity of the landslide evolution process and have higher prediction accuracy [5,17].

Advances in machine learning provide a powerful tool for numerical landslide model research. Zhou et al. [17] used kernel extreme learning for landslide displacement prediction. Zhu et al. [18] proposed a least squares support vector model and applied it to prediction of the Shuping landslide. Among them, Recurrent Neural Networks (RNNs) have particular advantages in dealing with sequential data [19,20]. Different from other neural networks, RNNs are the deepest algorithms [21], and they can effectively process data information with higher dimensions [22]. As a variant of RNNs, Long Short Term Memory (LSTM) networks perform better at storing and transferring historical information

than RNNs [23–26]. The utility of the LSTM in landslide research has been confirmed by many scholars [27–30]. Thus, we choose an LSTM network for landslide displacement prediction in this paper.

The Attention Mechanism (AM) is currently a powerful deep learning toolkit [31]. AM is similar to the human visual observation mechanism that can transfer key information from the input information [32]. AM has been successfully applied in several tasks, such as natural language processing [31], translation [33], and image recognition [34]. Li et al. [35] added the Attention Mechanism to the LSTM model and successfully realized the prediction of personal mobility. Ding et al. [36] proposed a spatio-temporal attention LSTM model for flood forecasting. Thus, we incorporate an Attention Mechanism with an LSTM neural network to capture significant variation and improve the model's performance.

Therefore, a novel model based on time series analysis and Attention Mechanism with Long Short Term Memory (AMLSTM) was proposed to predict landslide displacement. The Baishuihe landslide in China, Hubei province, is utilized for the experiment area. First, we use the Complete Ensemble Empirical Mode Decomposition with Adaptive Noise (CEEMDAN) algorithm to divide the total displacement into the trend term, the periodic term, and the residual term. By analyzing the corresponding relationship between displacement and external factors, a multiple factors AMLSTM model, is applied to predict the displacement, and it is compared with a further four machine learning models. A series of contrastive analyses are conducted to evaluate the performance of all of the models. The results indicate that the proposed CEEMDAN-AMLSTM model performs best in the experiment.

## 2. GNSS Time Series Analysis

### 2.1. Landslide Evolution Analysis

The evolution of landslides is the result of the interaction of geological conditions and external factors [37]. The non-linear and non-stationary landslide displacement series are particularly complex and changeable. Therefore, it is necessary to decompose the landslide time series and forecast each component separately. The corresponding time series of the landslide displacement can be expressed by the additive model:

$$y_i = T_i + S_i + R_i \tag{1}$$

where $y_i$ is the cumulative displacement, $T_i$ is the trend term, $S_i$ is the period term, and $R_i$ is the residual term.

### 2.2. Decomposition of Displacement Time Series

Many approaches have been recognized as being powerful tools for decomposing landslide displacement time series, and they include moving average [38], wavelet analysis [39], Variational Mode Decomposition (VMD) [40], and Empirical Mode Decomposition (EMD) [41]. The EMD method is an adaptive method that is used to analyze non-linear signals [42]. However, the model mixing problem constitutes an obstacle when using EMD. To address this problem, the Complete Ensemble Empirical Mode Decomposition with Adaptive Noise (CEEMDAN) method has been proposed in recent years [43]. Compared to the more commonly used EMD method, it has a better separation effect and is noise free. It has many applications in the fields of biological signal processing [44] and engineering [45], but its application in the geological field still needs to be explored.

The CEEMDAN decomposes the complex signal into a finite number of Intrinsic Mode Functions (IMFs). The basic process of the CEEMDAN is as follows [46]:

1. White Gaussian noises is added onto the lines of EEMD. The first IMF can be expressed as:

$$IMF_1 = \sum_{i=1}^{n} \frac{E_1(x + \varepsilon w_i)}{n} \tag{2}$$

where n is the number of decomposition, x is the original signal, $\varepsilon$ is a fixed coefficient, $w_i$ is the noise, and $E(\cdot)$ is the decomposition operator.

2.  The first residual, $r_1$, is calculated:

$$r_1 = x - IMF_1 \tag{3}$$

3.  For k = 2,3 ... , K, the $IMF_k$ and the kth residual can be calculated by:

$$IMF_k = \sum_{i=1}^{n} \frac{E_1(r_{k-1} + \varepsilon E_{k-1}(w_i))}{n} \tag{4}$$

$$r_k = r_{k-1} - IMF_k \tag{5}$$

4.  The process is calculated until the last residual, R, does not have more than two extrema points; the original signal can be expressed as:

$$x = \sum_{k=1}^{K} IMF_k + R \tag{6}$$

## 3. Attention Mechanism—LSTM Foresting Framework

### 3.1. LSTM

Long Short Time Memory (LSTM) was proposed by Hochreiter and Schmidhuber in 1997 [23]. The LSTM can learn information through a well-designed structure called a "gate". The gate can store and control the flow of information so that the state of the previous time step can be transferred to the next time step. The LSTM algorithm has three gates—update gate, forget gate, and output gate—to protect and control the cell state explosion in training [25]. The internal structure of the unit memory is as shown in Figure 1.

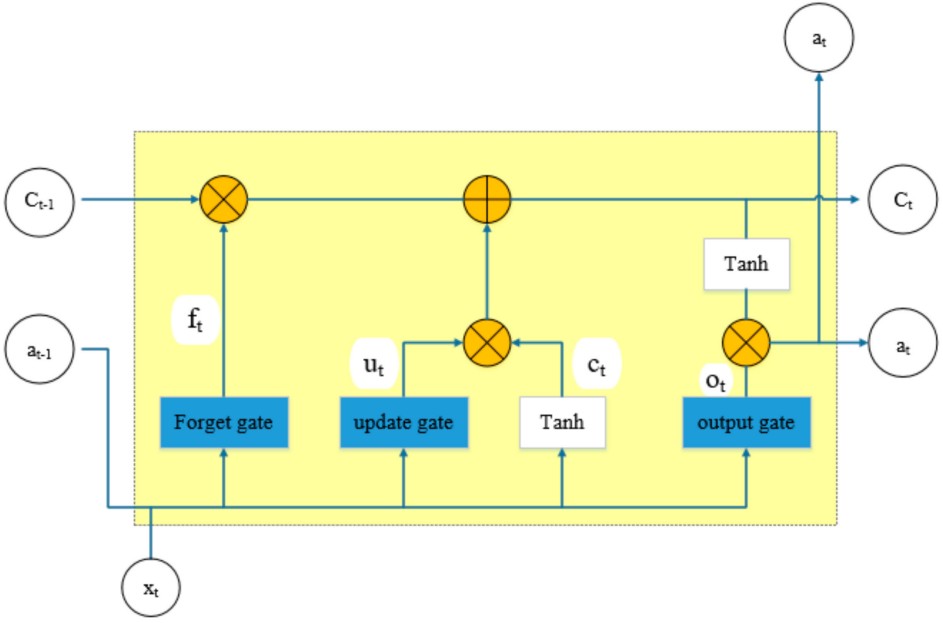

**Figure 1.** The internal structure of the Long Short Time Memory (LSTM) unit memory.

The $\otimes$ represents the element-wise product and $\oplus$ is the element-plus product. The forget gate represents how much of the previous moment unit sate, $c_{t-1}$, is retained by the current moment, $c_t$. The input gate determines how much of the current moment input, $x_t$, is saved in the unit state, $c_t$. The output gate controls how much of the unit state, $c_t$, is transferred to the output value, $h_t$, of the LSTM.

Equations (7)–(12) show the calculation process of LSTM:

$$f_t = \sigma\left(W_f * [a_{t-1}, x_t] + b_f\right) \tag{7}$$

$$u_t = \sigma(W_u * [a_{t-1}, x_t] + b_u) \tag{8}$$

$$\widetilde{c}_t = tanh(W_c * [a_{t-1}, x_t] + b_c) \tag{9}$$

$$c_t = f_t * c_{t-1} + u_t * \widetilde{c}_t \tag{10}$$

$$o_t = \sigma(W_o * [a_{t-1}, x_t] + b_o) \tag{11}$$

$$a_t = o_t * \tanh(c_t) \tag{12}$$

where $f_t$, $u_t$, and $o_t$ are gating vectors that respectively store the forgotten, updated, and output information of the storage unit memory; $c_t$ is the vector for the cell state; $a_t$ is the hidden state vector; $\sigma$ is the sigmoid function; and $x_t$ is the input vector. $W_f$, $W_u$, $W_c$, and $W_o$ are linear transformation matrices whose parameters need to be learned, and $b_f$, $b_u$, $b_c$, and $b_o$ are corresponding bias vectors.

Through the connection of several unit memories, the information flow can be transferred as shown in Figure 2.

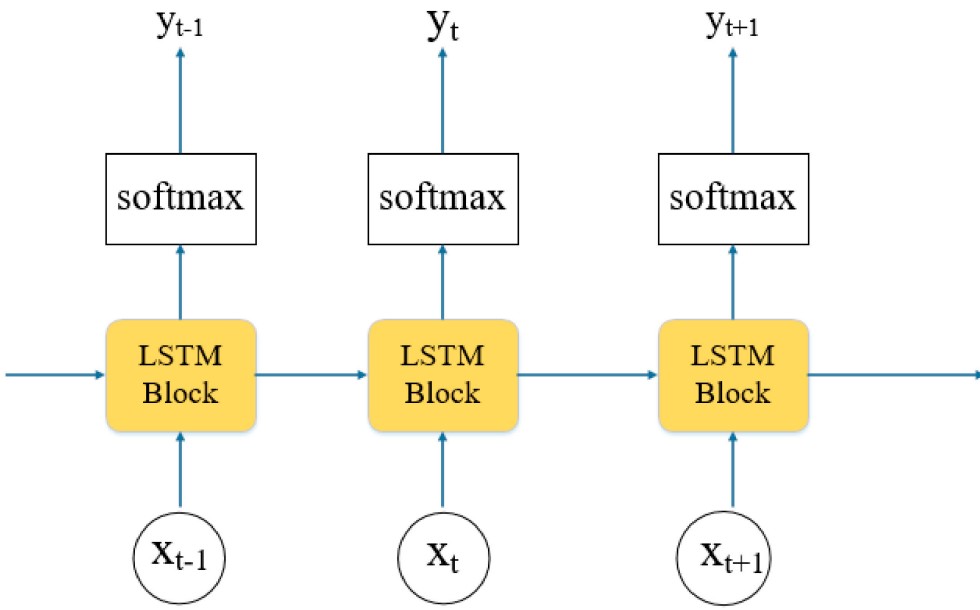

**Figure 2.** The workflow of LSTM.

### 3.2. Attention Mechanism

The Attention Mechanism is based on the visual Attention Mechanism found in human observation [32]. This mechanism helps the model focus on the salient information. The schematic of the Attention Mechanism layer is illustrated in Figure 3. The purpose of the attention layer is to enable the model to pay more attention to the significant information. Raffel et al. [47] proposed a reduced Feed-Forward Attention model, which was calculated as follows:

$$score_t = v(a_t) \tag{13}$$

$$w_t = \frac{\exp(score_t)}{\sum_{k=1}^{T} \exp(score_t)} \tag{14}$$

$$s = \sum_{t=1}^{T} w_t * a_t \tag{15}$$

where the score is the attention score, a is the state vector, v is the learnable function, w is the weight, and s is the context vector.

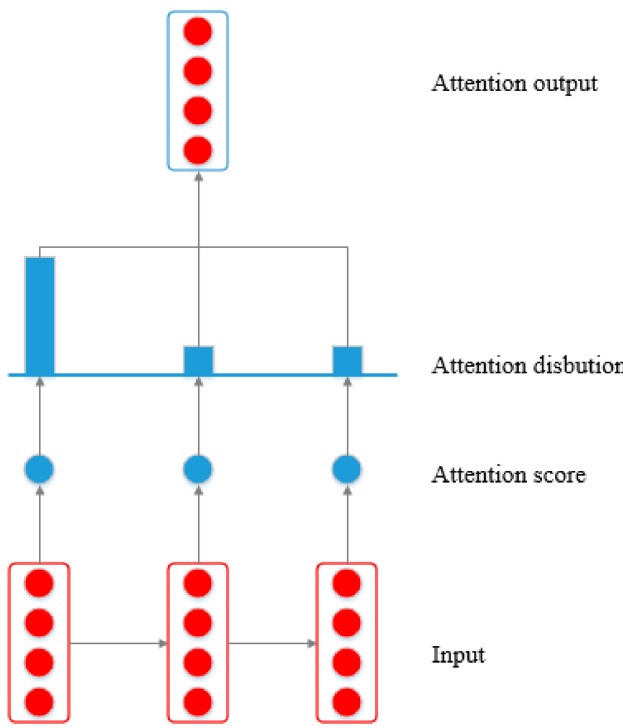

**Figure 3.** Schematic of the Attention Mechanism.

### 3.3. Attention Mechanism—LSTM Model

Based on the previous discussion, this paper applied the Attention Mechanism with LSTM (AMLSTM) model for landslide displacement prediction. The AMLSTM model includes an input vector, LSTM hidden layers, an attention layer, a fully connected layer, and output predicted values. The architecture of the AMLSTM model is shown in Figure 4.

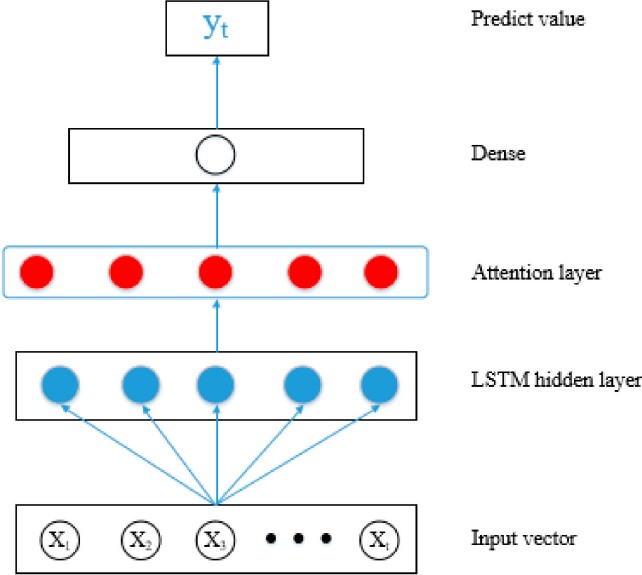

**Figure 4.** Architecture of the Attention Mechanism with LSTM Neural Network (AMLSTM NN).

### 3.4. Prediction Process with the Proposed Model

The basic flow of the proposed CEEMDAN-AMLSTM model is shown in Figure 5. Firstly, the landslide cumulative displacement is decomposed into three components: the trend term, the periodic term, and the residual term. The three terms are then predicted separately. The trend displacement is expressed as a monotone increasing function under the influence of internal geological factors. The prediction of the trend term can be carried out by fitting the growth curve with the univariate AMLSTM model. During the construction of the model, the displacement time series is put into the model only. The periodic displacement fluctuates under the influence of two external triggers: rainfall and reservoir water level. Therefore, a multivariable AMLSTM model is established and used to predict the periodic term. Three time series, the historical periodic displacement, rainfall, and reservoir water level are put into the model. Furthermore, the residual displacement affected by random factors shows smooth fluctuation function. The univariate AMLSTM model is adopted for the prediction work.

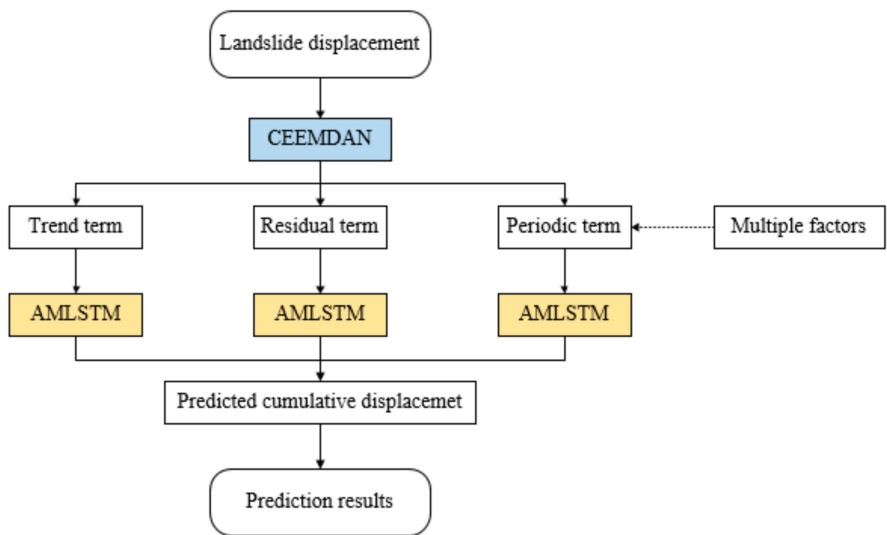

**Figure 5.** The architecture of the Complete Ensemble Empirical Mode Decomposition with Adaptive Noise (CEEMDAN)-AMLSTM model for landslide displacement prediction.

In the prediction experiments, the majority dataset is used to train the model. The original time series should be normalized and reshaped to meet the requirements of the model. After the AMLSTM model is constructed, the prediction ability is tested and demonstrated with the rest of the dataset.

Ultimately, the cumulative prediction displacement is obtained by adding the trend, the periodic, and the residual prediction displacements. The prediction results should be compared with the actual value to verity the performance.

### 3.5. Evaluation of Model Accuracy

Quantitative analysis were carried out to access the performance of the model. Three criterions—Root Mean Square Error (RMSE), Mean Absolute Error (MAE), and $R^2$—were employed to evaluate the prediction work. These metrics are described as follows:

$$\text{RMSE} = \sqrt{\frac{I}{N} * \sum_{i=1}^{N}(y_i - \hat{y}_i)^2} \tag{16}$$

$$\text{MAE} = \frac{1}{N} * \sum_{i=1}^{N}|y_i - \hat{y}_i| \tag{17}$$

$$R^2 = 1 - \frac{\sum_{i=1}^{N}(y_i - \hat{y}_i)^2}{\sum_{i=1}^{N}(y_i - \overline{y})^2} \tag{18}$$

where $y_i$ is the measured value, $\hat{y}_i$ is the prediction value, and $\overline{y}$ is the average value.

## 4. Experiment and Results

### 4.1. Study Area

The experimental area is located in Baishuihe, Zigui County, the Three Gorges Reservoir area of the Yangtze River in China. The Baishuihe landslide is located on the south bank of the Yangtze River, with a longitude of 110°32′09″ and a latitude of 31°01′34″ (Figure 6a). The slope is located on the south bank of the Yangtze River, spreading towards the Yangtze River in a ladder shape. The elevation of the back edge of the landslide is 410 m, bounded by the rock-soil boundary, and the front edge is about 70 m. It has been submerged below the reservoir water level. The east and west sides are bounded by bedrock ridges, and the overall slope is about 30°. The length of the north-south direction is 600 m, the width of the east-west direction is 700 m, the average thickness of the sliding body is about 30 m, and the volume is $1.26 \times 10^7$ m$^3$. Six Global Navigation Satellite System (GNSS) deformation monitoring points were installed on the surface of the landslide to form three longitudinal monitoring profiles (Figure 6b). The displacement was monitored once a month. Figure 7 shows the calculated displacement results from December 2006 to December 2012.

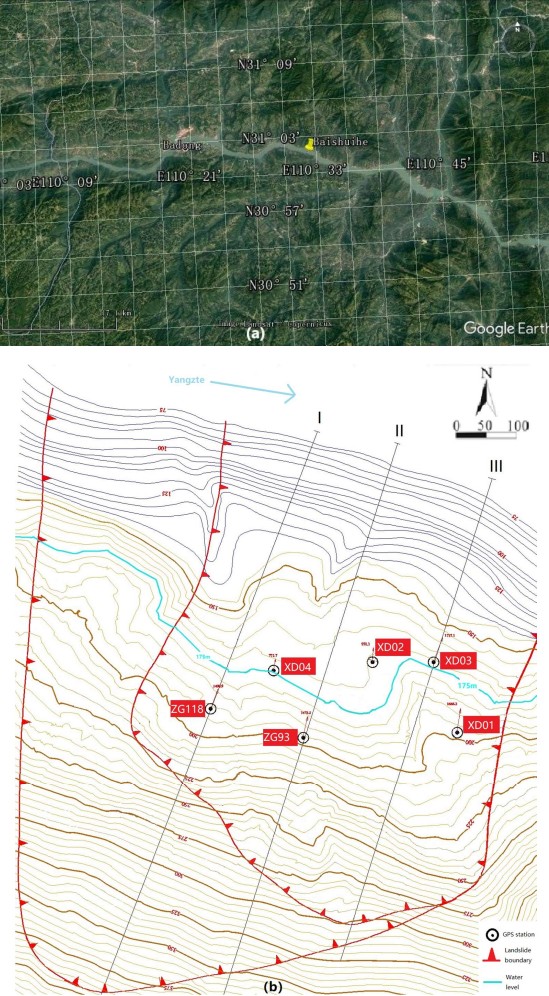

**Figure 6.** (**a**) Location map from Google Earth and (**b**) Locations of the monitoring Global Navigation Satellite System (GNSS) stations on the landslide.

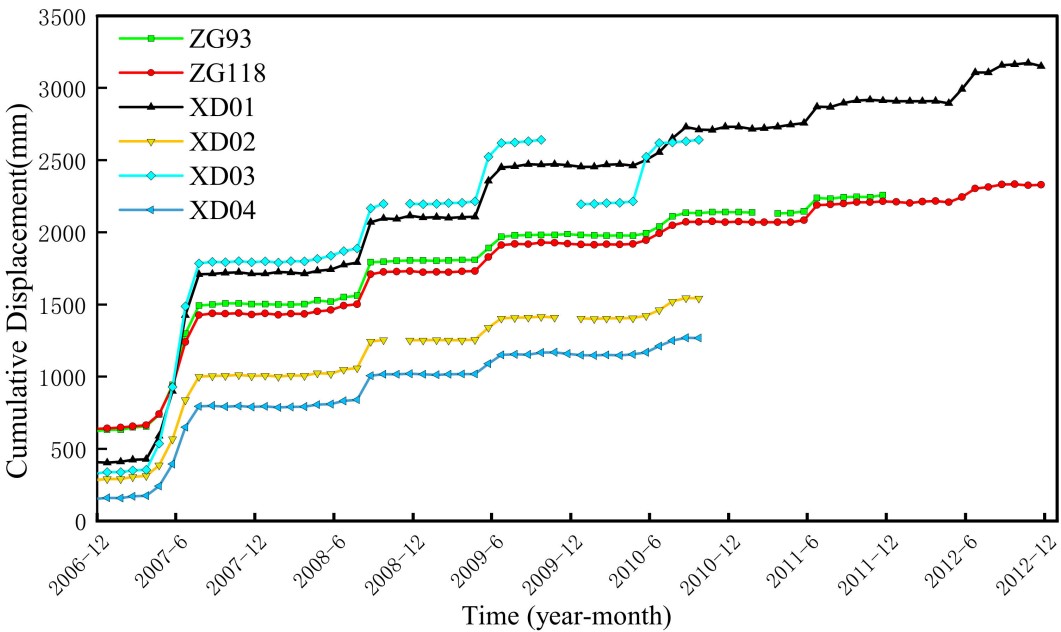

**Figure 7.** Cumulative displacement monitoring data based on six GNSS points.

It can be seen from Figure 7 that the landslide deformation is characterized by stepwise progressive creep deformation, and the landslide is still in the energy accumulation stage, showing a slow creep deformation state. In this experiment, ZG118 and XD01, the two points with the most abundant dataset, are selected for the prediction work. The measurements from December 2006 to November 2011 are used for training and the measurements obtained from November 2011 to November 2012 are used for testing. Each time interval of the train and test dataset is one month. The cumulative displacements, the reservoir water level, and the rainfall are plotted in Figure 8.

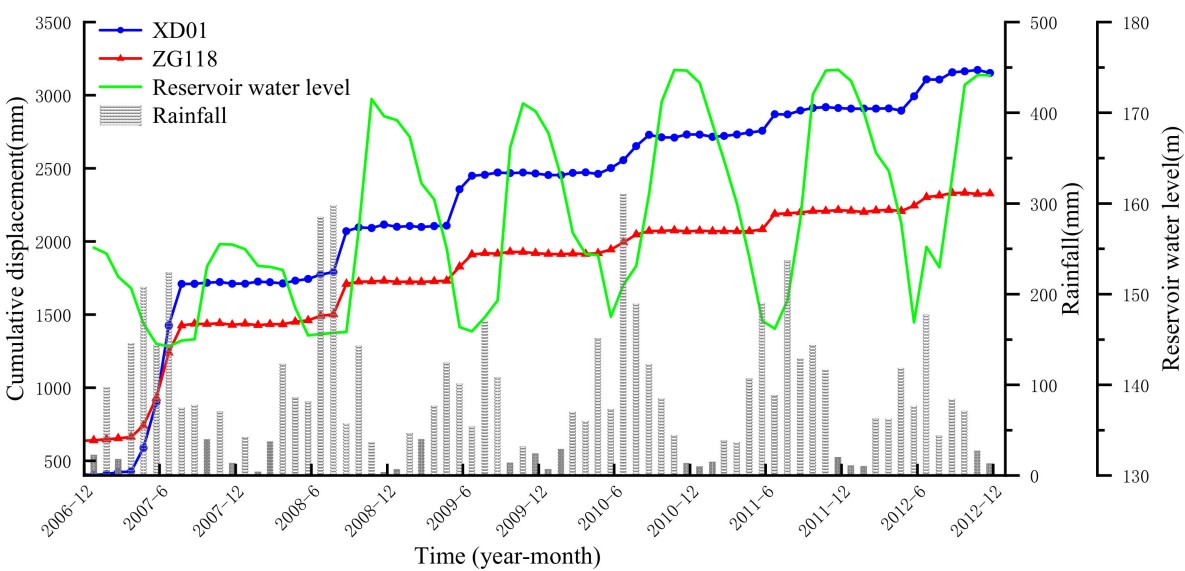

**Figure 8.** Relationship between rainfall, reservoir water level, and landslide displacement on ZG118 and XD01.

Figure 8 shows that the external periodic rainfall and reservoir water level both have an important influence. The displacement of XD01 and ZG118 increased significantly during a period of drastic decrease of the reservoir water level. For example, from May 2009 to July 2009, the reservoir water level dropped from 160 m to 145 m, and their periodic displacement increased by 200~300 mm, presenting a large step. In addition, heavy rain

also had an important effect on landslide displacement fluctuations. For example, from August 2008 to September 2008, the reservoir water level basically did not change but, due to the occurrence of 300 mm of heavy rain during this period, the landslide also showed a large deformation of 200 mm. Therefore, the reservoir water level and rainfall are considered to be the trigger factors of the Baishuihe landslide, leading to the occurrence of the periodic term displacement.

### 4.2. GNSS Time Series Analysis

According to landslide analysis theory, the cumulative displacement can be decomposed into trend displacement, periodic displacement, and residual displacement using the CEEMDAN algorithm. The results are as follows (Figure 9):

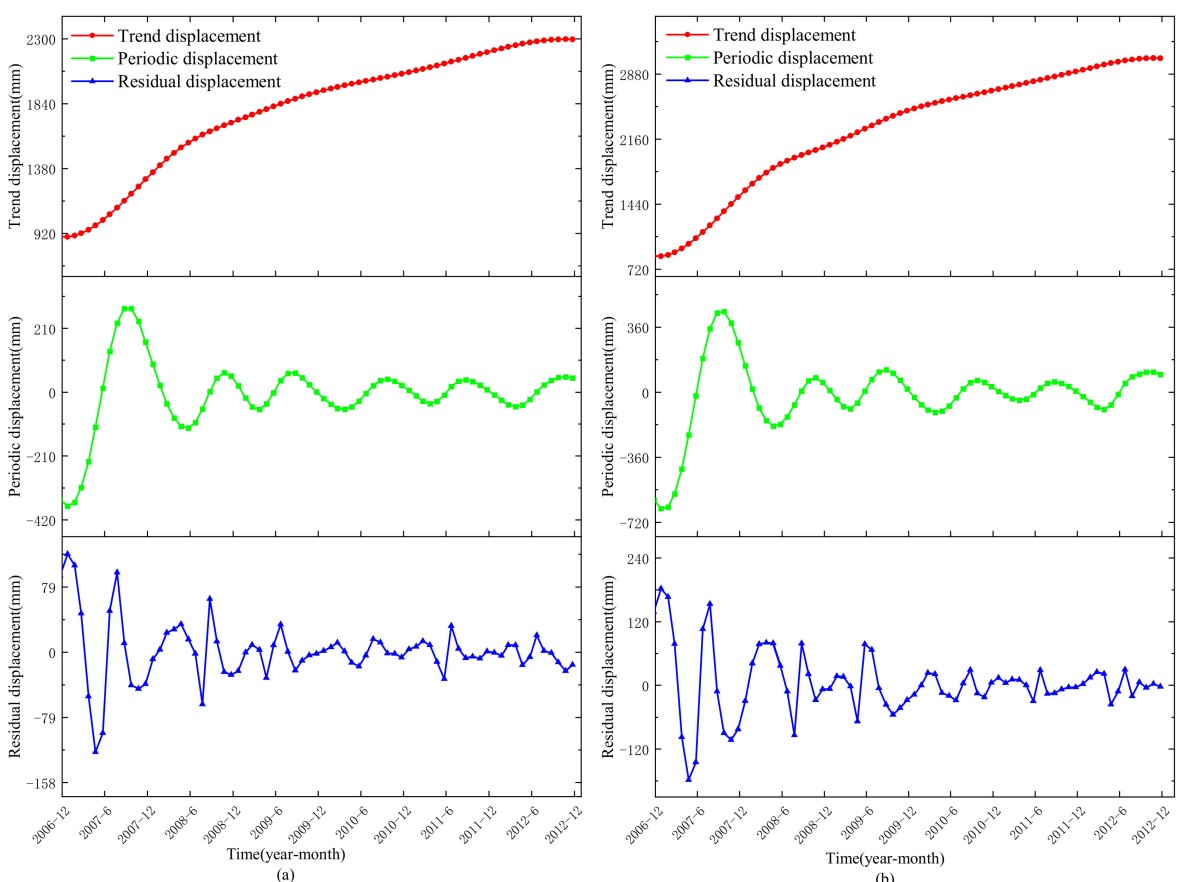

**Figure 9.** Three decomposed terms of the GNSS time series: (**a**) ZG118, (**b**) XD01.

### 4.3. Displacement Prediction

#### 4.3.1. Trend Displacement Prediction

Trend displacement is driven by geological conditions. Therefore, the univariate AMLSTM NN model is used to predict the trend displacement. In order to verify the validity of the proposed model, the experiment will be benchmarked with LSTM, Random Forest(RF), RNN, and Support Vector Machine(SVM). The prediction results of the test dataset are shown in Figure 10.

It can be seen in Figure 10 that the trend displacement of the ZG118 and XD01 points represent a smooth monotonically properties. The prediction work by the SVM shows the worst, and the prediction values of the AMLSTM, LSTM, RNN, and RF models show high agreement with the measured true value. The relative error analysis in Table 1 indicates that the AMLSTM, LSTM, and RF have excellent performance in trend term prediction work.

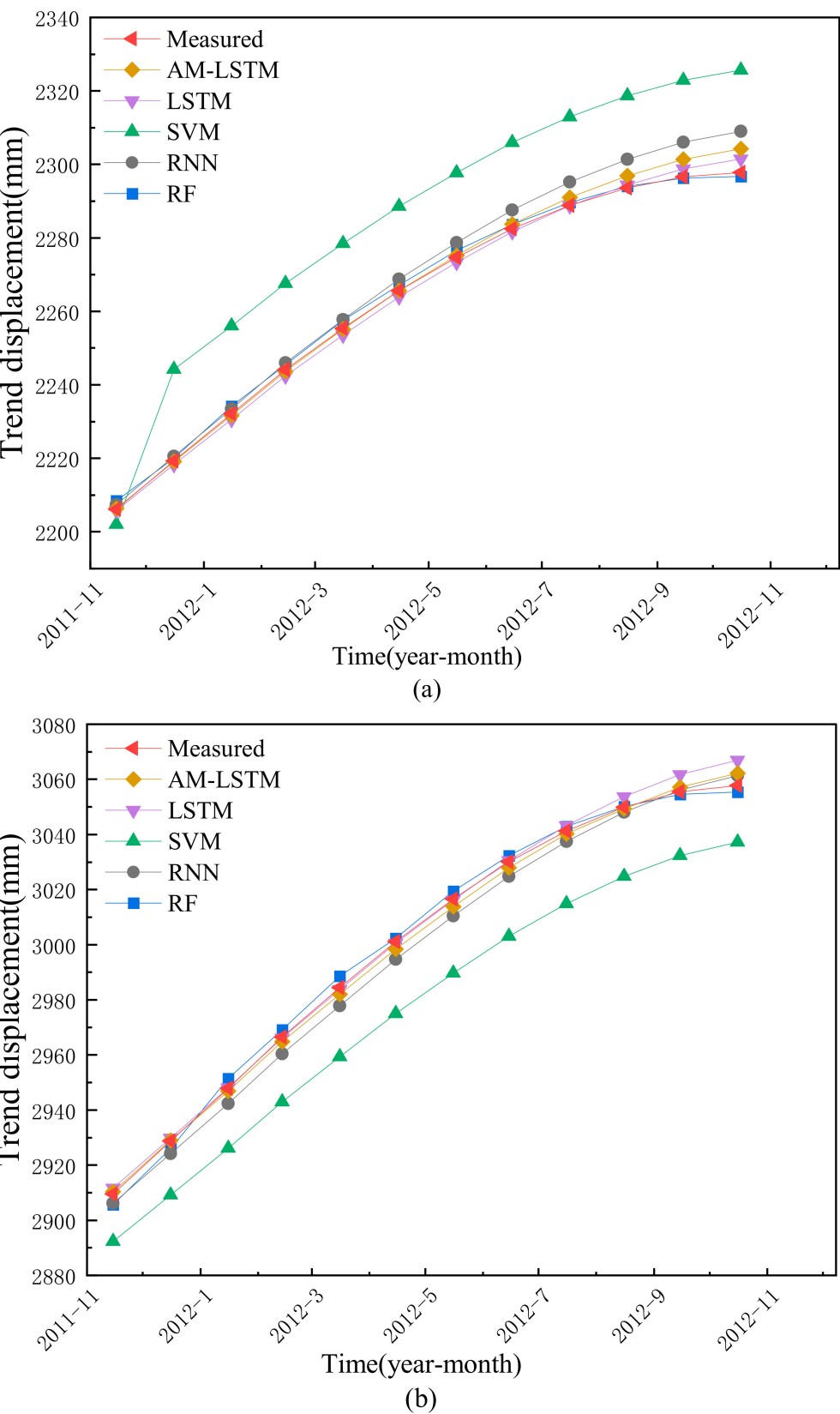

**Figure 10.** The prediction results of trend displacement by different methods: (**a**) ZG118, (**b**) XD01.

**Table 1.** The accuracy assessment of trend displacement by different prediction models.

| Model | RMSE | | MAE | | R² | |
|---|---|---|---|---|---|---|
| | ZG118 | XD01 | ZG118 | XD01 | ZG118 | XD01 |
| AMLSTM | 2.6152 | 2.1254 | 1.6785 | 1.7849 | 0.9925 | 0.9981 |
| LSTM | 1.6773 | 3.5006 | 1.4072 | 2.2426 | 0.9969 | 0.9949 |
| RNN | 5.6158 | 4.8276 | 4.5776 | 4.4758 | 0.9655 | 0.9904 |
| SVM | 23.3985 | 23.7356 | 22.6717 | 23.5418 | 0.4018 | 0.7678 |
| RF | 1.4897 | 2.6540 | 1.3317 | 2.3943 | 0.9976 | 0.9971 |

### 4.3.2. Periodic Displacement Prediction

Periodic term is a key component for displacement prediction. According to the analysis in Section 4.1, the external periodic rainfall and reservoir water level both have an important influence. In this section, the periodic displacement will be predicted by the multivariate AMLSTM, and the multivariate LSTM, the SVM, the RF, and the RNN are used as benchmarks. The predictive periodic displacements by the five models are shown in Figure 11 and Table 2.

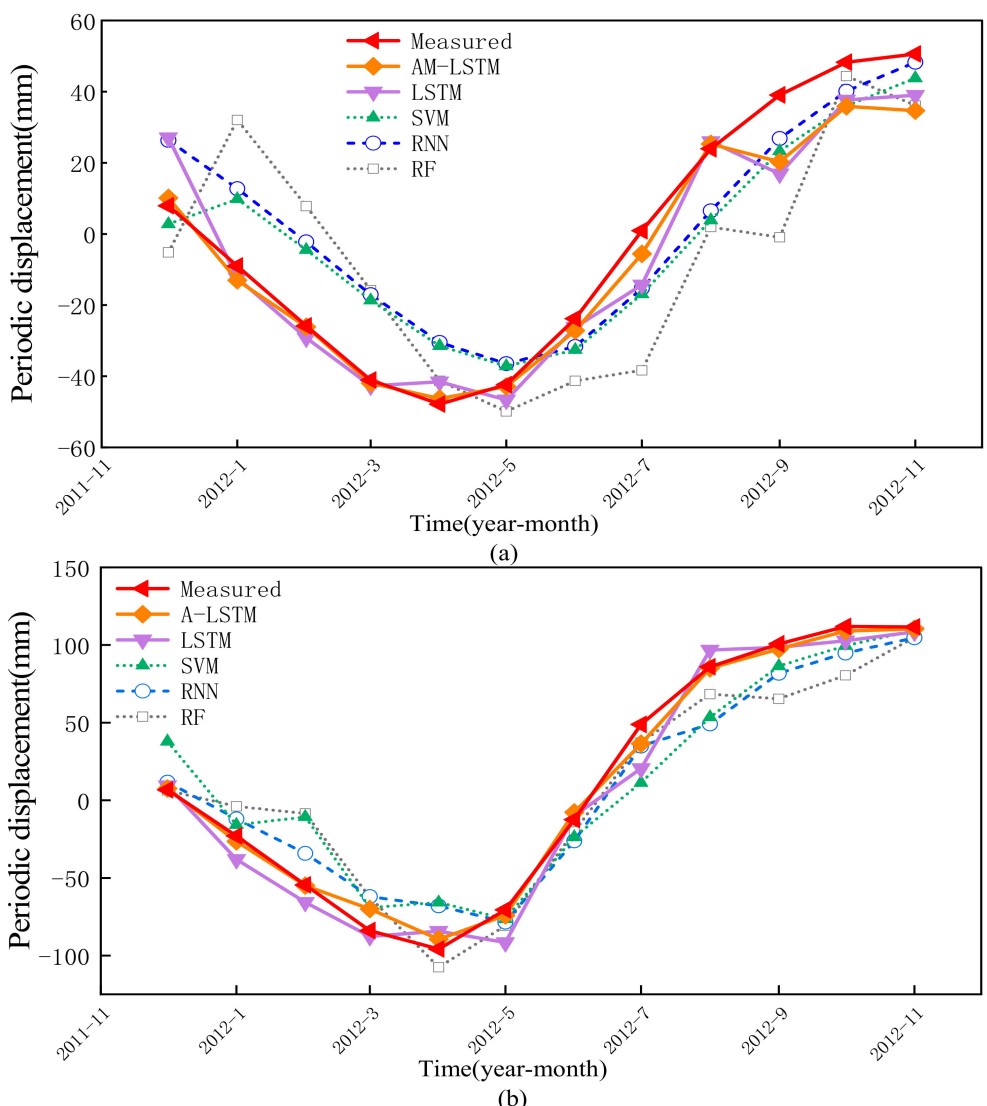

**Figure 11.** The prediction results of periodic displacement by different methods: (**a**) ZG118, (**b**) XD01.

**Table 2.** The accuracy assessment of periodic displacement by different prediction models.

| Model | RMSE | | MAE | | R$^2$ | |
|---|---|---|---|---|---|---|
| | **ZG118** | **XD01** | **ZG118** | **XD01** | **ZG118** | **XD01** |
| AMLSTM | 8.3714 | 6.1623 | 5.6456 | 4.5016 | 0.9404 | 0.9933 |
| LSTM | 10.9127 | 12.8428 | 8.5083 | 10.1266 | 0.8987 | 0.9711 |
| RNN | 16.1422 | 18.9561 | 14.5892 | 16.7908 | 0.7784 | 0.9371 |
| SVM | 15.4854 | 24.2245 | 14.236 | 20.2412 | 0.796 | 0.8972 |
| RF | 25.6368 | 22.3304 | 22.0298 | 18.298 | 0.441 | 0.9126 |

As shown in Figure 11, the predictions of the AMLSTM and LSTM methods are clearly better than the others, and the quantitative analysis suggest that the AMLSTM achieved the best performance, along with RMSE, MSE, and R$^2$, in periodic displacement prediction.

### 4.3.3. Residual Displacement Prediction

Traditionally, the residual term can be regarded as the noise, which is removed during the decomposition procedure. Throughout the test, the residual term does not belong to the white noise. Therefore, the prediction work of this term is necessary. In this experiment, the univariate AMLSTM, LSTM, SVM, RF, and RNN models are used to predict the residual displacement prediction.

Compared with the trend and the periodic term, the residual term is harder to adopt in a model because of its random characteristic. As shown in Figure 12 and Table 3, the AMLSTM offers a better prediction effect than the other four models.

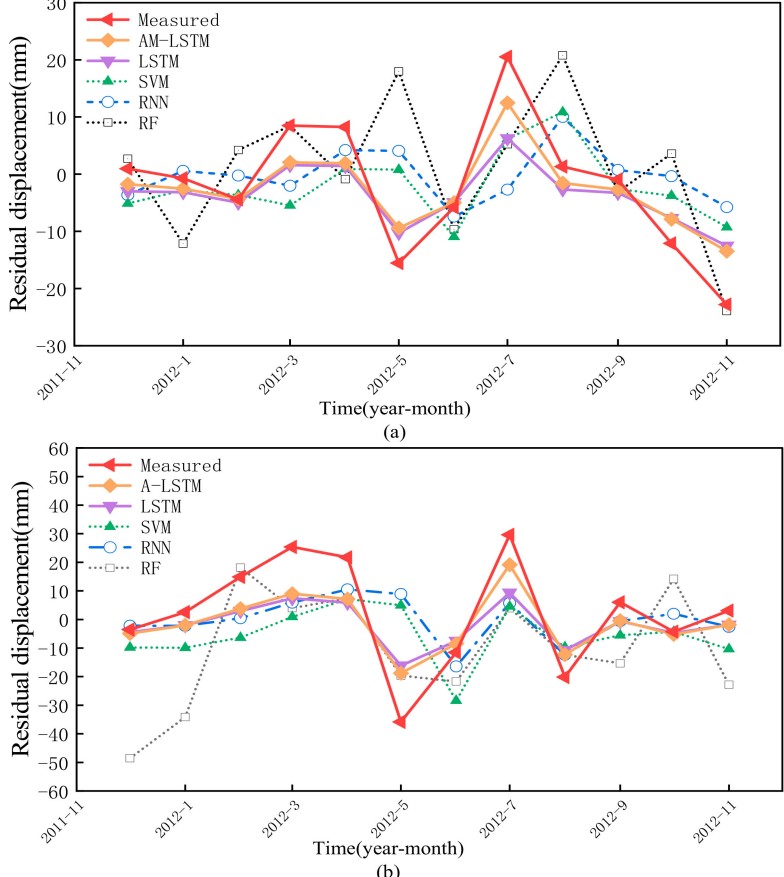

**Figure 12.** The prediction results of residual displacement by different methods: (**a**) ZG118, (**b**) XD01.

**Table 3.** The accuracy assessment of residual displacement by different prediction models.

| Model | RMSE | | MAE | | R² | |
|---|---|---|---|---|---|---|
| | ZG118 | XD01 | ZG118 | XD01 | ZG118 | XD01 |
| AMLSTM | 5.1002 | 9.8401 | 4.2185 | 8.2213 | 0.7897 | 0.7132 |
| LSTM | 6.4204 | 11.9279 | 5.1768 | 9.7219 | 0.6667 | 0.5785 |
| RNN | 11.5546 | 17.0916 | 9.0241 | 12.5840 | −0.0796 | 0.1346 |
| SVM | 9.7371 | 19.2705 | 8.2718 | 16.4355 | 0.2333 | −0.1001 |
| RF | 13.8302 | 23.4540 | 10.1748 | 20.5233 | −0.5467 | −0.6296 |

### 4.3.4. Total Displacement Prediction

The predicted cumulative displacements can be obtained by taking the sum of the trend, period, and residual displacements. The results are shown in Figure 13 and Table 4.

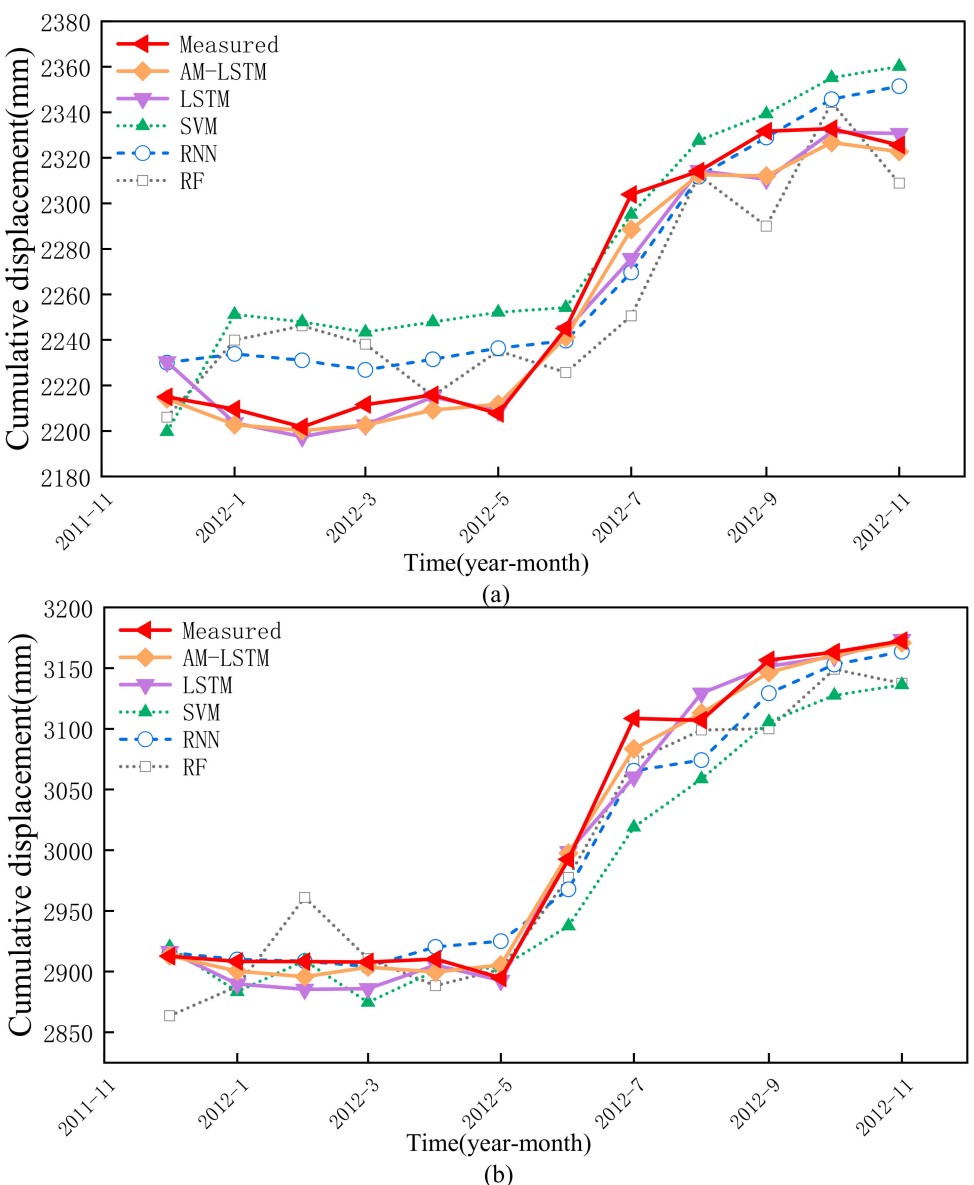

**Figure 13.** The prediction results of cumulative displacement by different methods: (**a**) ZG118, (**b**) XD01.

**Table 4.** The accuracy assessment of cumulative displacement by different prediction models.

| Model | RMSE | | MAE | | $R^2$ | |
|---|---|---|---|---|---|---|
| | ZG118 | XD01 | ZG118 | XD01 | ZG118 | XD01 |
| AMLSTM | 8.5514 | 10.249 | 6.5395 | 8.0242 | 0.9748 | 0.9918 |
| LSTM | 11.7059 | 18.8873 | 7.8044 | 13.3813 | 0.9528 | 0.9723 |
| RNN | 20.4623 | 21.4569 | 17.6515 | 16.3575 | 0.8556 | 0.9634 |
| SVM | 29.1695 | 41.3469 | 25.6171 | 33.3799 | 0.7066 | 0.8673 |
| RF | 28.5883 | 32.0225 | 23.5398 | 26.5033 | 0.7182 | 0.9204 |

The results show that, although some of the prediction values slightly deviate from the real measured data, the AMLSTM model shows the best performance, because this model not only considers multiple external factors, but also optimizes the LSTM algorithm by adding an attention layer. It can better reflect the response relationship between displacement and trigger factors. Moreover, the cumulative displacements are predicted badly by the SVM and RF models.

From a quantitative point of view, the RMSE and MAE of the AMLSTM model are lower than the LSTM, RNN, SVM, and RF models. These results reveal that the AMLSTM shows the most stable prediction performance. Secondly, the $R^2$ of the AMLSTM are higher than the others. The results indicate that the AMLSTM model has done the best accuracy prediction work. Therefore, the superiority of the AMLSTM can be proved.

## 5. Conclusions

The traditional landslide prediction model directly deletes the residual items. Moreover, most classic deep learning prediction models do not highlight the impact of important information on the results, so they cannot accurately predict the displacement. This paper used the CEEMDAN and the Attention Mechanism, combined with the LSTM NN to establish a dynamic prediction model for landslide displacement prediction. To corroborate its feasibility and applicability, the proposed model was applied to the Baishuihe landslide area, and joint multiple impact factors were considered here for prediction. By comparing to the prediction effects of other models, the prediction accuracy demonstrated a competitive performance. The results strongly suggest the effectiveness and feasibility of the AMLSTM model in landslide displacement prediction. This novel CEEMDANAM-LSTM strategy can be recommended to other landslide prediction works and has great potential in landslide risk assessment.

**Author Contributions:** Conceptualization, J.W. and G.N.; methodology, J.W.; software, J.W.; validation, S.W. and X.R.; formal analysis, S.G.; investigation, G.N. and S.G.; writing—original draft preparation, J.W.; writing—review and editing, J.W. and H.L. All authors have read and agreed to the published version of the manuscript.

**Funding:** This study was financially supported by the National Key Research and Development Scheme Strategic International Cooperation in Science and Technology Innovation Program, grant number: 2018YFE0206500. National Program on Key Basic Research Project (973 Program), grant number: 2013CB733205.

**Institutional Review Board Statement:** Not applicable.

**Informed Consent Statement:** Not applicable.

**Data Availability Statement:** Not applicable.

**Acknowledgments:** The dataset we used in this paper includes the GNSS time series, rainfall and reservoir water level data set of Baishuihe landslide provided by Chinese National Cryosphere Desert Data Center (http://www.crensed.ac.cn/portal/, accessed on 9 February 2021). The authors acknowledge Google Earth for providing the map and Origin software. Thanks to the editor Aguero Gui and the anonymous reviewers.

**Conflicts of Interest:** The authors declare no conflict of interest.

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
