# Peer review of "Landslide Deformation Prediction Based on a GNSS Time Series Analysis and Recurrent Neural Network Model"

_remotesensing, doi:10.3390/rs13061055_

Round 1
Reviewer 1 Report
You are using Fig.10 (Page 10) as a prove for statement (lines 209-211): "It can be seen in Fig.10 that the trend displacement of the ZG118 and XD01 points are predicted accurately by the AM-LSTM and LSTM models, whereas the SVM, RNN and RF cannot predict well. I agree with the statement regarding RF model, but it is not clear from the Fig.10 for SVM and RNN models. Therefore I propose to authors either to modify the picture showing clearly the difference or to explain it in the text!
Author Response
Dear Reviewer,
Thanks very much for taking your time to review this manuscript. We really appreciate your comments and suggestions!
We go through Fig.10 (Page 11) carefully and find it difficult to identify. We are trying to modify the picture. But we find it difficult to distinguish the line from each other because the prediction values are especially similar. Then we calculate the RMSE, MAE and R2 value of each models. A new Table 1 is added in Page 11 Line 271. And we explain it in Page 12 Line 277-282.
A revised document with the correction portion using "Tracking changes" function is attached and we hope meet with approval.
We are very grateful to your comments for the manuscript. Should you have any questions, please contact us without hesitate.

Reviewer 2 Report
In this work, the Authors show the development of a novel model, Attention Mechanism with Long Short Time Memory Neutral Network (AMLSTM NN) based on Complete Ensemble Empirical Mode Decomposition with Adaptive Noise (CEEMDAN) in order forecast displacement in the case study of Baishuihe landslide. Moreover, in order to validate the model the resulta have been compared to other statistical algorithms (SVM, RNN, RF).
The study provides a good description of the used approach characterized by a simple and clear structure. While the analyses carried out would seem too concise. A more detailed statistical study could be useful in comparing the results obtained by the different approaches also for the different components (trend, periodic and residual). In the paper this analysis is present only in the final part. It could be useful to introduce the statistical analysis already in the introduction stating the methodology.
It is an original contribution, appropriate material for Remote Sensing journal. For these reasons, I suggest publishing the paper after minor revision.
Some other comments:
- Check the references cited in the text (Pg. 2 line 53; Pg. 2 line 55….)
- I propose to make the title of section 2.2 explicit
- Caption of figure 2: workflow instead flow
- Figure 5: please insert the North symbol, the grid with the coordinates and the scale bar.
- Pg 6 line 160: change Fig. 5b with 5a
- Figure 6: which component is reported (North, East, Up)? Are they values projected on the slope? Could the direction of displacement be included in Figure 5b?
- the citation of the figures in the text is preferable to put it in full i.e. Figure XX
Please note that there are many typing errors. English language revision is advised
Author Response
Dear Reviewer,
Thank you very much for your good evaluation and kind comments concerning our manuscript. Those comments are valuable and helpful for revising and improving our paper. We have studied comments carefully and have made extensive modification on the original manuscript. A revised document with the correction portion using "Track Changes" function is attached and we hope meet with approval. The main corrections in the paper and the responds to the comments are as follows:
Point 1: A more detailed statistical study could be useful in comparing the results obtained by the different approaches also for the different components (trend, periodic and residual). In the paper this analysis is present only in the final part. It could be useful to introduce the statistical analysis already in the introduction stating the methodology.
Response 1: Thanks for the kind suggestion. The statistical analysis part has been moved to Page 6 Line 178-189 as a new section 3.5. And the analysis of the trend, periodic and residual term has been added in Page 11 Line 271 Table 1, Page 12 Line 294 Table 2 and Page 13 Line 323 Table 3.
Point 2: Check the references cited in the text (Pg. 2 line 53; Pg. 2 line 55….)
Response 2: We are very sorry for our mistakes. The cited format of references [35] and [36] have been corrected in Pg.2 line 56 and line58.
Point 3: I propose to make the title of section 2.2 explicit
Response 3: The title of section 2.2 has been modified to “Decomposition of displacement time series” in Page 2 Line 81.
Point 4: Caption of figure 2: workflow instead flow
Response 4: The word “flow” has been changed to “workflow” in Figure 2 (Page 4 Line 131).
Point 5: Figure 5: please insert the North symbol, the grid with the coordinates and the scale bar
Response 5: We have modified Figure 6a (Page 7 Line 205) with the North symbol, the grid and the scale bar.
Point 6: Pg 6 line 160: change Fig. 5b with 5a
Response 6: We changed Fig. 5b with 6b in Page 7 Line 203.
Point 7: Figure 6: which component is reported (North, East, Up)? Are they values projected on the slope? Could the direction of displacement be included in Figure 5b?
Response 7: The displacement showed in Figure 6 (now Figure 7) is the total displacement; Because the landslide slope varies from 10° to 30°, it’s hardly to obtain the accurate slope angle of the point. So the values are not projected on the slope; The direction of displacement is changed in the measurements once a month. So we couldn’t add the direction in Figure 6b.
Point 8: the citation of the figures in the text is preferable to put it in full i.e. Figure XX
Response 8: We are very sorry for our inappropriate writing. The citation is corrected in Page 4 Line 135, Page 5 Line 149, Page 7 Line 195 and 203, Page 8 Line 217, Page 9 Line 225, 230, Page 10 Line 264, Page 12 Line 277 and 290, Page 14 Line 353 and 358.

Reviewer 3 Report
The manuscript “Landslide deformation prediction based on GNSS time series analysis and recurrent neural network model” presents a novel displacement prediction model applied to monitoring data acquired by permanent GNSS receivers installed in an active landslide.
Authors present the comparison of the predictive performance of the newly developed “Attention Mechanism with Long Short Time Memory Neutral Network” (AMLSTM NN) based on “Complete Ensemble Empirical Mode Decomposition with Adaptive Noise” (CEEMDAN) with respect to other predictive models (LSTM, RF, RNN and SVM). AMLSTM NN results appear to model accurately the two displacement time-series ingested in the model. Although, from my side Authors should present more in detail the dataset preparation for the analysis:
- Did the dataset have been organized in training and test portions?
- Which are the model inputs? Displacements time-series only?
In the attached pdf minor correction are noted.

Author Response
Dear Reviewer,
Thank you very much for the review and the valuable comments on our manuscript. We have carefully revised the manuscript and answered the questions according to the suggestions. All the revisions of the article are red marked. Because of your suggestions, the revised article becomes better and readers can get more valuable information. We sincerely hope this manuscript will be finally accepted. .
Point 1: Did the dataset have been organized in training and test portions
Response 1: Yes, the dataset had been normalized and reshaped in the actual programming. We are very sorry for our negligence. We added a new section 3.4 “Prediction process with the proposed model” in Page 6 Line 155-174 to explain the data processing.
Point 2: Which are the model inputs? Displacements time-series only?
Response 2: In Section 4.3.1 and 4.3.3, only the displacements time-series put into the prediction model. But in section 4.3.2, three time-series displacement, rainfall and reservoir water level are put into the prediction model. We added the explain text in Page 6 Line 155-174.
Point 3: Page 1 Line 14 ingest?
Response 3: The word “recognize” had been replaced with “ingest” in Page 1 Line 14.
Point 4: Page 8 Line 184-185 In case Did you thought on computing a chi-square test in order to determine independence between rainfall and reservoir level?
Response 4: We have carried out a chi-square test between rainfall and reservoir level. The results show that they are dependent upon each other. Although they are dependent, the influence on the landslide are different. The historical data showed that there were almost no deformations on the Baishuihe landslide before 2003 water storage in the Three Gorges Reservoir area. But the surface cracks appeared frequently after the increase of reservoir level. Meanwhile, field investigations showed that the surface cracks appeared frequently after heavy rainfall. Therefore, rainfall and reservoir level were trigger factors for Baishuihe landslide. Both of them were considered in the prediction work.
Point 5: Page 10 Figure 10 typo error
Response 5: The type error “Displacement” has been corrected in Figure 10
Point 6: Page 12 Line 235 typo error
Response 6: The type error “estimable” has been changed to a new word “adopt” in Page 14 Line 352.

Round 2
Reviewer 3 Report
Dear Authors,
Thank for addressing my previous comments. Now the rationale of your methodology is clear to me.
On regard to the newly written section 3.4 "Prediction process with the proposed model": while I do understand the sense of it, I would recommend to improve the english language.
Moreover, you should be more accurate and explicit in presenting your training and test dataset at lines 168-171: please define each time interval of the training and test datasets.
Author Response
Dear Reviewer,
Thanks very much for taking your time to review this manuscript. We really appreciate all your comments and suggestions! We have uploaded the revised manuscript with all the changes highlighted by using the track changes mode in MS Word. The main corrections in the paper and the responds to the comments are as follows:
Point 1: On regard to the newly written section 3.4 "Prediction process with the proposed model": while I do understand the sense of it, I would recommend to improve the english language
Response 1: Thanks for the kind suggestion. We rewrote the Section 3.4 from Line 147 to Line 163, Page 6. We have tried our best to improve the English language.
Point 2: Moreover, you should be more accurate and explicit in presenting your training and test dataset at lines 168-171: please define each time interval of the training and test datasets
Response 2: Thank you for your detailed work. We added the description of train and test dataset in Page 9 Line 201 to 203 as follows: “The measurements from December 2006 to November 2011 are used for training. And the measurements obtained from November 2011 to November 2012 are used for testing. Each time interval of the train and test dataset is a month.”
